# Polydopamine Doping and Pyrolysis of Cellulose Nanofiber Paper for Fabrication of Three-Dimensional Nanocarbon with Improved Yield and Capacitive Performances

**DOI:** 10.3390/nano11123249

**Published:** 2021-11-30

**Authors:** Luting Zhu, Kojiro Uetani, Masaya Nogi, Hirotaka Koga

**Affiliations:** SANKEN (The Institute of Scientific and Industrial Research), Osaka University, 8-1 Mihogaoka, Ibaraki 567-0047, Osaka, Japan; uetani@eco.sanken.osaka-u.ac.jp (K.U.); nogi@eco.sanken.osaka-u.ac.jp (M.N.)

**Keywords:** polydopamine doping, cellulose nanofiber, pyrolysis, 3D porous nanocarbon, supercapacitor

## Abstract

Biomass-derived three-dimensional (3D) porous nanocarbons have attracted much attention due to their high surface area, permeability, electrical conductivity, and renewability, which are beneficial for various electronic applications, including energy storage. Cellulose, the most abundant and renewable carbohydrate polymer on earth, is a promising precursor to fabricate 3D porous nanocarbons by pyrolysis. However, the pyrolysis of cellulosic materials inevitably causes drastic carbon loss and volume shrinkage. Thus, polydopamine doping prior to the pyrolysis of cellulose nanofiber paper is proposed to fabricate the 3D porous nanocarbons with improved yield and volume retention. Our results show that a small amount of polydopamine (4.3 wt%) improves carbon yield and volume retention after pyrolysis at 700 °C from 16.8 to 26.4% and 15.0 to 19.6%, respectively. The pyrolyzed polydopamine-doped cellulose nanofiber paper has a larger specific surface area and electrical conductivity than cellulose nanofiber paper that without polydopamine. Owing to these features, it also affords a good specific capacitance up to 200 F g^−1^ as a supercapacitor electrode, which is higher than the recently reported cellulose-derived nanocarbons. This method provides a pathway for the effective fabrication of high-performance cellulose-derived 3D porous nanocarbons.

## 1. Introduction

There has been rapid progress in the fabrication and design of three-dimensional (3D) porous nanocarbon materials because they provide several advantages such as high surface areas, short diffusion spaces for fast reaction kinetics, and efficient electron pathways [1,2]. As a result, 3D porous nanocarbons have been actively investigated for energy storage applications [3,4,5], including supercapacitors [6,7] and batteries [8,9,10].

The majority of carbon-based materials have been conventionally fabricated using petroleum-based precursors [11]. For example, carbon nanofibers have been produced from polyacrylonitrile, pitches, and phenolic resins [12], while commercial carbon fibers, including carbon nanofibers, are produced from petroleum-based precursors only [13], more than 96% of commercial carbon fibers are made from polyacrylonitrile [13,14]. From the viewpoint of sustainable development, there has been an increased demand to replace these non-renewable petroleum-based precursors with abundant and renewable precursors [15,16]. Therefore, renewable biomass-derived 3D porous nanocarbons have been actively developed [17,18,19,20].

Cellulose, which is directly produced from natural plants, is known as the most abundant and renewable carbohydrate polymer on earth [21]. Cellulose intrinsically exists in the plant cell wall [22], and it can be extracted in the form of nanofibers by physical and/or chemical methods [23,24,25]. Because plant-derived cellulose nanofibers can be fabricated into paper with 3D porous nanostructures by the solvent exchange and papermaking processes [26,27], cellulose nanofiber paper may be a promising precursor to the fabrication of 3D porous nanocarbons by pyrolysis. Cellulose possesses the theoretical carbon content of 44.4 wt%, which refers to the residual six carbon atoms per anhydroglucose unit in the cellulose molecular structure [28]. However, the pyrolysis of cellulose is inevitably accompanied by combustion reactions, forming volatile low-molecular-weight carbon-containing substances such as CO, CO_2_, alcohols, and ketones and resulting in low yield and drastic volume shrinkage [28,29]. Thus, it is a challenge to fabricate cellulose-derived 3D porous nanocarbons with improved yield and volume retention.

Thus, in this study, polydopamine doping prior to the pyrolysis of the cellulose nanofiber paper is proposed. The small amount of dopamine, which is a well-known biomolecule that is present in various animals [30] is polymerized in situ in the cellulose nanofiber suspension, and it is then fabricated into paper with 3D porous nanostructures. The resulting polydopamine-doped cellulose nanofiber paper is pyrolyzed to prepare the 3D porous nanocarbon with improved yield and volume retention compared to that pyrolyzed without polydopamine doping. Furthermore, polydopamine doping also yields cellulose nanofiber papers with enhanced specific surface areas and electrical conductivity, thereby providing high capacitive performances for energy storage applications.

## 2. Materials and Methods

### 2.1. Materials

Cellulose nanofiber water suspension (BiNFi-s cellulose, raw material: softwood bleached kraft pulp) was obtained from Sugino Machine Ltd., Toyama, Japan. Dopamine hydrochloride, *tert*-butyl alcohol, tris(hydroxymethyl)aminomethane, potassium hydroxide, and 0.5 M hydrochloric acid solution were obtained from Nacalai Tesque, Inc., Kyoto, Japan. All chemicals were of analytical reagent grade and were used without further purification.

### 2.2. In Situ Polymerization of Dopamine in Cellulose Nanofiber Suspension

In situ polymerization of dopamine in the cellulose nanofiber suspension was performed according to a previous report [31]. Briefly, 0.24 g tris(hydroxymethyl)amino- methane was first added to cellulose nanofiber water suspension (0.2 wt%, 200 mL), and the pH was adjusted to 8.0 by means of a 0.5 M hydrochloric acid solution. Then, different amounts of dopamine hydrochloride (0.025 g, 0.05 g, and 0.1 g) were mixed with the suspension, and the suspension was then stirred for 1 day at approximately 25 °C. The suspension turned black in color during the dopamine polymerization [32,33].

Neat polydopamine was also prepared by adding 0.24 g tris(hydroxymethyl)amino- methane to 200 mL distilled water and by adjusting the pH to 8.0 with 0.5 M hydrochloric acid followed by adding 0.1 g dopamine hydrochloride and stirring the mixture for 1 day at approximately 25 °C. Subsequently, the suspension was centrifugated at 10,000 rpm for 15 min (Model 7000, Kubota Corporation Co., Ltd., Tokyo, Japan). The precipitate was washed with distilled water, and it was then centrifuged in the same conditions; this washing treatment was repeated three times. Finally, the polydopamine was dispersed in distilled water and frozen overnight in a refrigerator (SJ-23T, Sharp Corp., Osaka, Japan) and freeze dried (EYELA FDU-2200, Tokyo Rikakikai Co., Ltd., Tokyo, Japan).

### 2.3. Preparation of Polydopamine-Doped Cellulose Nanofiber Papers

The aqueous mixture of polydopamine and cellulose nanofibers was dewatered by suction filtration (KST-47, Advantec Toyo Kaisha, Ltd., Tokyo, Japan) through a commercial membrane filter (H020A090C, hydrophilic polytetrafluoroethylene membrane with pore size of 0.2 μm, Advantec Toyo Kaisha, Ltd., Tokyo, Japan). Then, 200 mL distilled water was gently added onto the wet sheet, and it was then vacuum filtrated to remove excess reagents. To form the 3D porous nanostructures [27,34], 200 mL of *tert*-butyl alcohol was further added, followed by vacuum filtration. The obtained wet paper was then peeled off from the filter, stored at approximately −18 °C for over 2 h in a refrigerator, and freeze-dried to prepare the polydopamine-doped cellulose nanofiber papers.

### 2.4. Pyrolysis of Polydopamine-Doped Cellulose Nanofiber Papers

The polydopamine-doped cellulose nanofiber papers were cut into square-shaped samples with the size of 1.5 × 1.5 cm^2^, followed by pyrolysis at 700 °C for 1 h under N_2_ gas using a desktop gas convertible vacuum furnace (KDF75, Denken-Highdental Co., Ltd., Kyoto, Japan). The heating and cooling rates were set at 2 °C min^−1^. The N_2_ flux was set at 0.5 L min^−1^. The weight and volume of the original and polydopamine-doped cellulose nanofiber papers were measured before and after pyrolysis. To calculate the weight and volume retention after pyrolysis, five pieces of the paper samples were prepared for each polydopamine content. The polydopamine particles were also pyrolyzed under the same conditions. To calculate the weight retention after pyrolysis, three batches of the polydopamine particles were prepared.

### 2.5. Electrochemical Tests as Supercapacitor Electrodes

The electrochemical tests as a supercapacitor electrode were performed by a three-electrode system using 6 M KOH aqueous electrolyte (ModuLab XM, Solartron Analytical-AMETEK Advanced Measurement Technology Inc., Berkshire, UK). Approximately 2–5 mg pyrolyzed polydopamine-doped cellulose nanofiber paper was directly evaluated as a working electrode, which was covered by nickel foam for electrical signal collection. A Hg/HgO electrode was used as the reference electrode. A platinum wire was used as the counter electrode. Galvanostatic charge/discharge tests and the cyclic voltammetry (CV) were operated at the current densities of 0.5−20 A g^−1^ and the scan rate of 10 mV s^−1^, respectively. Electrochemical impedance spectroscopy (EIS) was measured in the frequency range from 0.1 to 100 kHz, with an amplitude of 5 mV. A single-electrode gravimetric capacitance (*C*, F g^−1^) was estimated by the following formula, according to charge/discharge curves:*C* = *IDt*/(*m*Δ*V*) (1)
where *I* and *Dt* are the charge/discharge current (A) and the discharge time (s), respectively, *m* is the weight (g) of the pyrolyzed polydopamine-doped cellulose nanofiber paper, and Δ*V* is the voltage change (*V*), which excluded the internal resistance drop in the discharge period.

### 2.6. Characterization

Thermogravimetric (TG) analyses were carried out under a nitrogen flux of 60 mL min^−1^ at a heating rate of 10 °C min^−1^ (TGA Q50N2, TA Instruments, New Castle, DE, USA), with approximately 15 mg sample placed in the platinum pan. The surface and cross-section observations were performed using field-emission scanning electron microscopy (FE-SEM) (SU-8020, Hitachi High-Tech Science Corp., Tokyo, Japan). The acceleration voltage was set at 2 kV. Before the FE-SEM observation, platinum sputtering was performed on the samples at 20 mA for 10 s. The nitrogen physisorption measurements were operated at 77 K (NOVA 4200e, Quantachrome Instruments, Kanagawa, Japan). Brunauer–Emmett–Teller (BET) analyses were carried out at relative pressures ranging from 0.01 to 0.3. X-ray diffraction (XRD) analysis was performed by means of an Ultima IV X-ray diffractometer (Rigaku Corp., Tokyo, Japan) with Ni-filtered Cu Kα radiation (1.5418 Å) at 40 kV and 40 mA, and the scanning angle (2θ) was from 5° to 80° at the scanning rate of 1° min^−1^. Raman spectra were recorded at a laser wavelength and a power of 532 nm and 0.1 mW, respectively (RAMAN-touch VIS-NIR-OUN, Nanophoton Corp., Osaka, Japan). Elemental analysis was operated by a JM10 instrument (J-Science Lab Co., Ltd., Kyoto, Japan). The volume resistivity measurements were performed by a four-probe resistivity meter (Loresta-GP, MCP-T610, Mitsubishi Chemical Analytech Co., Ltd., Tokyo, Japan); three pieces of the pyrolyzed polydopamine-doped cellulose nanofiber papers were prepared for each polydopamine content, and three different positions were measured for each paper sample.

## 3. Results

### 3.1. Pyrolysis of Polydopamine-Doped Cellulose Nanofiber Paper

Polydopamine doping and the pyrolysis of the cellulose nanofiber paper were performed as shown in Figure 1. In brief, different amounts of dopamine were first added to the cellulose nanofiber suspension for in situ polymerization to polydopamine. Then, the black-colored aqueous dispersion of the polydopamine-doped cellulose nanofibers was dewatered by suction filtration and solvent exchange, and the samples were then freeze dried. As such, cellulose nanofiber papers with different polydopamine contents (3.4–8.2 wt%) were obtained (Appendix A) and pyrolyzed at 700 °C.

As shown in Figure 2a, the original and polydopamine-doped cellulose nanofiber papers shrank and became brittle to some degree after pyrolysis, but the papers were free-standing, which enabled them to be handled easily for characterization and application testing. Notably, polydopamine doping improved the volume retention (from 15.0% to 23.6%) (Figure 2b), weight retention (from 8.6% to 14.6%) (Figure 2c), and carbon yield (from 16.8% to 28.9%) (Figure 2d, see also Appendix A) of the pyrolyzed cellulose nanofiber paper. Moreover, the weight retention and carbon yield of the pyrolyzed polydopamine-doped cellulose nanofiber paper were higher than the estimated ones. For example, the weight retention and carbon yield of the 4.3 wt% polydopamine-doped cellulose nanofiber paper after pyrolysis were 13.0% and 26.4%, respectively, which were higher than those (10.9% and 19.8%, respectively) estimated from the original cellulose nanofiber paper and neat polydopamine after pyrolysis. Because the polydopamine content was low, it can be expected that the increased weight retention and carbon yield are mainly due to the weight retention of cellulose.

To explain the improved weight and volume retention by polydopamine doping, the TG and derivative thermogravimetric (DTG) curves were analyzed. As it can be seen in Figure 2e, the thermal decomposition temperature (5% weight decrease in the TG curves [35]) of the original cellulose nanofiber paper increased after polydopamine doping from 268.3 °C (polydopamine: 0 wt%) to 282.8 (3.4 wt%), 279.3 (4.3 wt%), and 277.8 °C (8.2 wt%), while that of neat polydopamine was 220.2 °C. The DTG_peak_ temperature and derivative weight of the cellulose nanofiber paper decreased when the polydopamine content increased (Figure 2f). These results suggested that polydopamine doping improves the thermal stability of the cellulose nanofiber paper, thereby affording the increased weight yield and volume. The improved thermal stability can be ascribed to the radical scavenging effect of the catecholic compounds (e.g., polydopamine [36] and melanin [37]). While free radicals, which are generated from cellulose during pyrolysis, unfavorably generate gaseous compounds to cause large weight loss [38], polydopamine can scavenge some of these free radicals and can suppress the weight loss.

### 3.2. 3D Porous Nanostructures of Pyrolyzed Polydopamine-Doped Cellulose Nanofiber Paper

The 3D porous nanostructures provide unique properties, including high surface areas and permeability, which are beneficial for a variety of applications including energy storage. Therefore, the 3D porous nanostructures of the original and the pyrolyzed polydopamine-doped cellulose nanofiber papers were analyzed accordingly (Figure 3). The undoped cellulose nanofiber paper with a thickness of approximately 270 μm contained a porous nanofiber network (Figure 3a) and a layered structure (Appendix A), comprising the 3D porous nanostructures. The occurrence of these 3D porous nanostructures can be ascribed to the solvent exchange from water to *tert*-butyl alcohol during the paper fabrication process because the *tert*-butyl alcohol with low surface tension suppressed the aggregation of the cellulose nanofibers upon drying [27].

Further, the 3D porous nanostructures were maintained even after polydopamine doping (Figure 3b–d and Appendix A). The polydopamine particles formed by in situ polymerization in the cellulose nanofibers had a diameter ≤~50 nm (Figure 3b–d), which was much smaller than the neat polydopamine nanoparticles prepared in the absence of cellulose nanofibers (Appendix A), suggesting that cellulose nanofibers can restrict the growth and aggregation of the polydopamine nanoparticles. Owing to the nanosized polydopamine particles, the polydopamine-doped cellulose nanofiber papers had a higher specific surface area (132–145 m^2^ g^−1^) than the original (94 m^2^ g^−1^) (Figure 3i).

It was also confirmed that the polydopamine-doped cellulose nanofiber papers maintained their 3D porous nanostructures after pyrolysis at 700 °C (Figure 3e–h and Appendix A). The thicknesses of the pyrolyzed polydopamine-doped cellulose nanofiber papers were approximately 108, 112, 129, and 142 μm at a polydopamine content of 0%, 3.4%, 4.3%, 8.2%, respectively. The N_2_ adsorption and desorption isotherms of the pyrolyzed polydopamine-doped cellulose nanofiber papers indicated an obvious increase of N_2_ adsorption in the low relative pressure range and a hysteresis loop in the high relative pressure range, suggesting the presence of micropores (<2 nm) and mesopores (2–50 nm, characteristic of type IV isotherm), respectively [39,40]. The 4.3 wt% polydopamine-doped cellulose nanofiber paper showed the highest specific surface area (617 m^2^ g^−1^) after pyrolysis, which was much larger than that without polydopamine doping (506 m^2^ g^−1^). However, at 8.2 wt% polydopamine, a lower specific surface area (510 m^2^ g^−1^) after pyrolysis was observed, suggesting that the excess amount of polydopamine decreases the resulting specific surface area. The appropriate amount of polydopamine nanoparticles within the cellulose nanofiber paper partially restrain the shrinkage of the porous nanofiber networks upon pyrolysis and provide higher volume retention and specific surface areas (see also Figure 2b), while an excess amount of polydopamine nanoparticles may block the porous nanostructures, thereby decreasing the specific surface areas. Our results indicate that 3D porous nanostructures with high specific surface areas can be obtained after the pyrolysis of appropriately polydopamine-doped cellulose nanofiber paper.

### 3.3. Molecular Structure and Electrical Conductivity of Pyrolyzed Polydopamine-Doped Cellulose Nanofiber Papers

The chemical structures and electrical properties of the pyrolyzed polydopamine-doped cellulose nanofiber papers were also analyzed (Figure 4). The Raman spectra showed the G band at approximately 1580 cm^−1^ and D band at approximately 1340 cm^−1^, which are ascribed to the graphitic carbon domains and the disordered graphitic carbon structures (e.g., edge of graphitic domains and in-plane imperfections) [41], respectively (Figure 4a). This indicates the formation of graphitic carbon structures with disordered regions, such as oxygen- and nitrogen-doped carbon structures, as a result of pyrolysis (see also Appendix A). The graphitic structures after pyrolysis were also confirmed by the XRD spectra. Broad peaks at approximately 23° and 43° (Figure 4b) were observed, which are assigned to the (002) and (10) bands of graphite, respectively [42,43]. The crystallite sizes in the in-plane (*L_a_*) and stacking (*L_c_*_)_ directions of the graphitic carbon can be estimated using Scherrer’s formula from the (10) and (002) lattice planes, respectively [41,43]. As shown in Figure 4c, the graphitic carbon domains increased their average width (*L_a_*) from ~1.9 to 2.4 nm (interplanar distance *da* = ~0.2 nm) with increasing polydopamine content, while their average thickness (*L_c_*) was almost constant at ~1.0 nm. Because neat polydopamine pyrolyzed at the same temperature (700 °C) showed a lower average width of ~1.7 nm than the pyrolyzed polydopamine-doped cellulose nanofiber papers, it was suggested that the combination of polydopamine and cellulose nanofibers can promote the growth of the graphitic carbon domains in the in-plane direction after pyrolysis. Such promoted growth of electrically conductive graphitic carbon domains also contributed toward the increased electrical conductivity (decreased volume resistivity); the volume resistivities of the pyrolyzed polydopamine-doped cellulose nanofiber papers decreased from 8.8 to 5.5 Ω cm when the polydopamine content increased (Figure 4d), indicating high electrical conductivity and the positive effect of polydopamine doping.

### 3.4. Application as an Electrode for a Supercapacitor

Tuning the porous structure and molecular structure of cellulose-derived nanocarbons by doping is effective in enhancing their energy storage performance [44,45,46]. To demonstrate the significance of the polydopamine doping and pyrolysis of the cellulose nanofiber paper, the pyrolyzed polydopamine-doped cellulose nanofiber paper was applied as a supercapacitor electrode in a 6 M KOH aqueous electrolyte with a three-electrode system. The previously reported cellulose-derived porous carbons frequently required additives, such as conductive carbon and binder [47,48,49]. On the other hand, our pyrolyzed polydopamine-doped cellulose nanofiber papers were successfully applied as conductive and free-standing electrodes. Such free-standing property can ensure sufficient transportation of electrons and electrolyte ions between the current collector and the electrodes [50]. The pyrolyzed polydopamine-doped cellulose nanofiber papers showed a larger surrounding area of the CV curve and longer discharge time in the galvanostatic charge/discharge curve than those without polydopamine doping, indicating the higher capacitance of the pyrolyzed polydopamine-doped cellulose nanofiber papers (Figure 5a,b). The specific capacitance values were plotted as a function of the current densities based on the charge-discharge curves (Figure 5c). Notably, the pyrolyzed 4.3 wt% polydopamine-doped cellulose nanofiber paper showed the highest specific capacitance of up to approximately 200 F g^−1^ at current densities of 0.5−20 A g^−1^, which was considerably higher than that without polydopamine doping (approximately 160 F g^−1^).

The mechanism that was responsible for the enhanced specific capacitance was discussed. Regardless of polydopamine doping, the pyrolyzed cellulose nanofiber papers presented a rectangular shape with distortion in the CV curves and slight nonlinear charge-discharge curves, which indicate an electric double-layer capacitor behavior with redox pseudocapacitance [47] (Figure 5a,b). The redox pseudocapacitance is derived from disordered carbon structures, such as oxygen- and nitrogen-containing groups [51], which can be estimated as the coulombic efficiency obtained from the ratio of the discharge duration to the charge duration in the charge-discharge curves [52]. The pyrolyzed cellulose nanofiber papers with polydopamine contents of 0, 3.4, 4.3, and 8.2 wt% showed coulombic efficiencies of 111.11, 114.64, 112.96, and 110.76% at 0.5 A g^−1^, respectively, indicating minimal difference in their redox pseudocapacitance. The resistance of electrolyte ion diffusion within the electrodes and the interfacial charge-transfer at the electrode-electrolyte interface was further analyzed by Nyquist plots, which consist of a linear line at low-frequency region and a semicircle at high-frequency region [47] (Figure 5d). The pyrolyzed cellulose nanofiber papers with and without polydopamine showed a nearly vertical line, representing efficient ionic diffusion [40], which is possibly due to their 3D porous nanostructures (Figure 3e–f). It has been reported that K^+^ (electrolyte ion) with a large radius can cause the structural deformation of the electrode during the insertion and extraction processes [53,54]. In this study, however, the pyrolyzed polydopamine-doped cellulose nanofiber paper maintained its porous nanostructures after electrochemical tests (Appendix A). Notably, the pyrolyzed polydopamine-doped cellulose nanofiber papers showed a smaller diameter of a semicircle than that without polydopamine doping. This indicates a more efficient interfacial charge-transfer at the electrode-electrolyte interface [40] for the polydopamine-doped samples, owing to their higher electrical conductivity as electrodes (Figure 4d). Thus, the highest specific capacitance of the pyrolyzed 4.3 wt% polydopamine-doped cellulose nanofiber paper is a result of its enhanced electrical conductivity and high specific surface area (Figure 3j). Moreover, the pyrolyzed polydopamine-doped cellulose nanofiber paper provided higher specific capacitance than previously reported cellulose- and other biomass-derived porous nanocarbon materials such as carbonized cellulose aerogel [46], wood-derived carbon nanofiber aerogel [49], and hybrid nanocellulose derived hierarchical porous carbon film [55], suggesting its great potential as promising electrodes in supercapacitors (Appendix A).

## 4. Conclusions

We have demonstrated that the polydopamine doping and pyrolysis of cellulose nanofiber paper can be a promising method that can be used to fabricate the 3D porous nanocarbons with improved carbon yield and volume retention. The pyrolyzed polydopamine-doped cellulose nanofiber paper also offered higher specific surface areas and electrical conductivity than that without polydopamine, thereby affording higher specific capacitance of up to 200 F g^−1^. The specific capacitance of the polydopamine-doped cellulose nanofiber paper was superior to the cellulose-derived nanocarbons that have been reported previously. Our results suggest that this method will facilitate the effective fabrication of high-performance cellulose-derived 3D porous nanocarbons for energy storage and other electronic applications. This method can be extended to other bionanomaterials, paving the way for future sustainable electronics.

## Figures and Tables

**Figure 1 nanomaterials-11-03249-f001:**
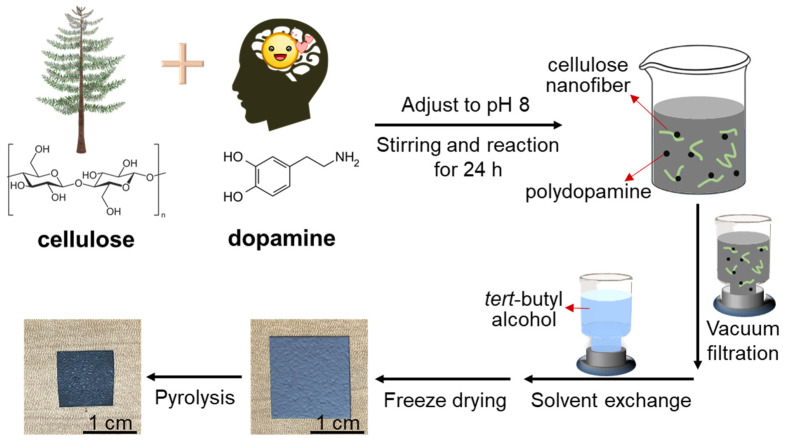
Schematic illustration of the sequential procedure for polydopamine doping and pyrolysis of the cellulose nanofiber paper.

**Figure 2 nanomaterials-11-03249-f002:**
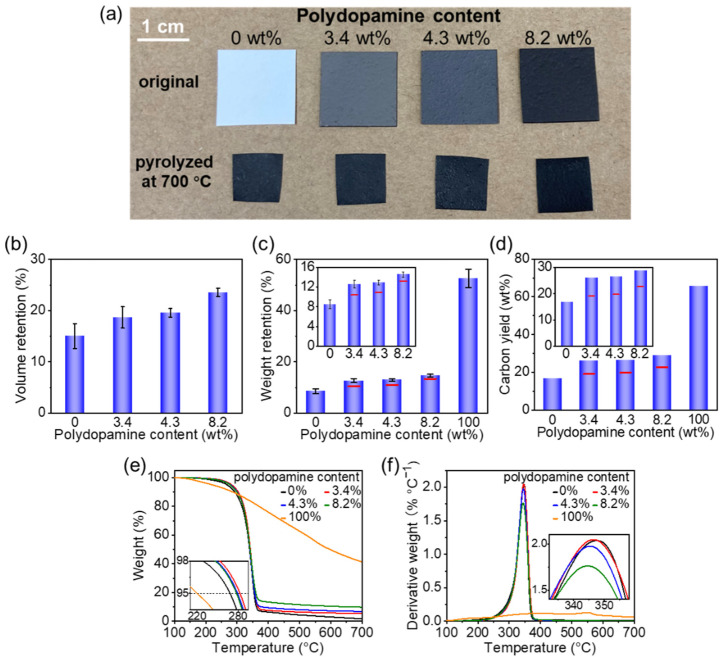
Pyrolysis of the polydopamine-doped cellulose nanofiber paper. (**a**) Cellulose nanofiber papers with different polydopamine contents before and after pyrolysis at 700 °C; (**b**) volume retention and (**c**) weight retention (the red lines indicate the weight retention estimated from the original cellulose and neat polydopamine) and (**d**) carbon yield (the red lines indicate the carbon yield estimated from the pyrolyzed cellulose and pyrolyzed polydopamine) of the pyrolyzed cellulose nanofiber papers with different polydopamine contents; (**e**) thermogravimetric and (**f**) derivative thermogravimetric curves of the cellulose nanofiber papers with different polydopamine contents and neat polydopamine.

**Figure 3 nanomaterials-11-03249-f003:**
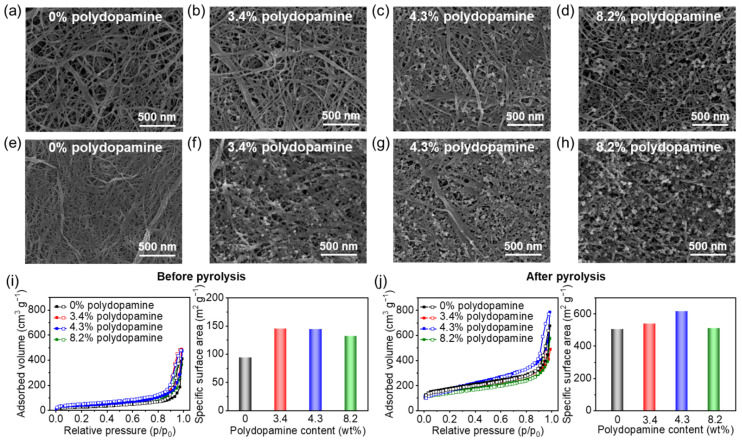
Three-dimensional porous nanostructures of pyrolyzed polydopamine-doped cellulose nanofiber papers. Field emission scanning electron microscopy images of the cellulose nanofiber papers with different polydopamine content (**a**–**d**) before and (**e**–**h**) after pyrolysis at 700 °C; N_2_ adsorption and desorption isotherms of cellulose nanofiber papers with different polydopamine content and the corresponding specific surface areas (**i**) before and (**j**) after pyrolysis at 700 °C.

**Figure 4 nanomaterials-11-03249-f004:**
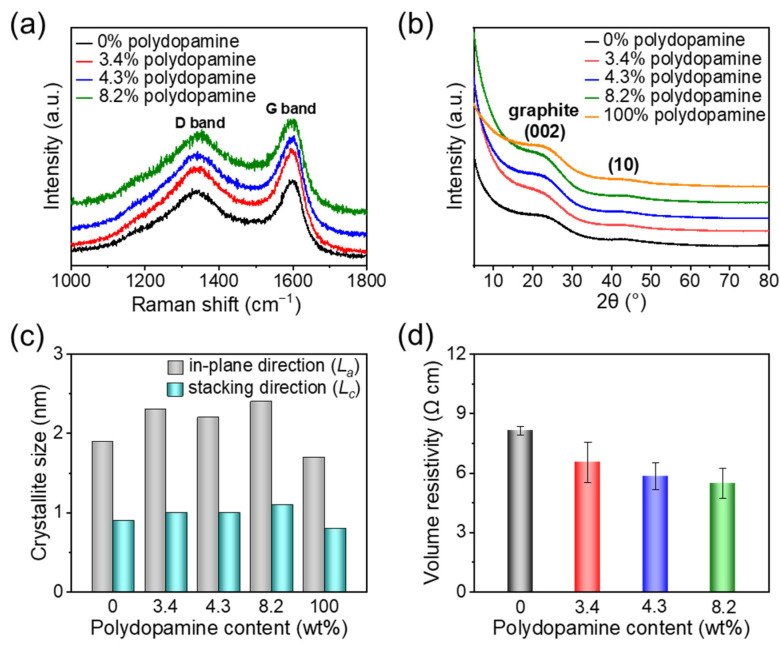
Molecular structure and electrical conductivity of the polydopamine-doped cellulose nanofiber papers pyrolyzed at 700 °C. (**a**) Raman spectra, (**b**) X-ray diffraction spectra, (**c**) crystallite sizes of the graphitic carbon domains in the in-plane (*L_a_*) and stacking (*L_c_*) directions, and (**d**) volume resistivity.

**Figure 5 nanomaterials-11-03249-f005:**
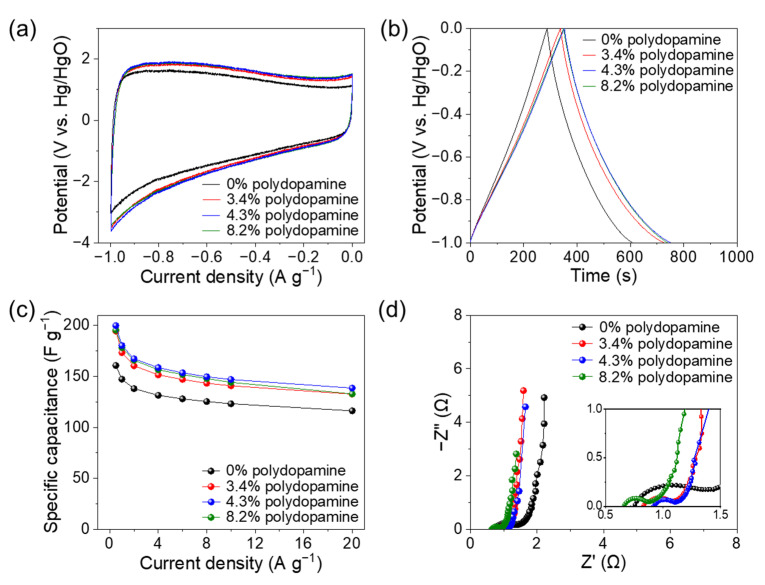
Electrochemical tests of the polydopamine-doped cellulose nanofiber papers pyrolyzed at 700 °C. (**a**) Cyclic voltammetry curves (scan rate: 10 mV s^−1^); (**b**) galvanostatic charge-discharge curves (current intensity: 0.5 A g^−1^); (**c**) specific capacitance at the current densities of 0.5−20 A g^−1^; (**d**) Nyquist plots.

## Data Availability

The data is included in the main text and/or the Appendix A.

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
