# Peer review of "Polydopamine Doping and Pyrolysis of Cellulose Nanofiber Paper for Fabrication of Three-Dimensional Nanocarbon with Improved Yield and Capacitive Performances"

_nanomaterials, 2021, doi:10.3390/nano11123249_

Round 1

Reviewer 1 Report

Manuscript ID: nanomaterials-1469503

Title: Polydopamine doping and pyrolysis of cellulose nanofiber paper for fabrication of three-dimensional nanocarbon with improved yield and capacitive performances

In this work, the authors designed three-dimensional nanocarbon by polydopamine doping prior to pyrolysis of cellulose nanofiber paper. They found that small amount of polydopamine (4.3 wt%) improves carbon yield and volume retention after pyrolysis at 700 °C from 16.8 to 26.4% and 15.0 to 19.6%, respectively. I suggest that this manuscript is very interesting. Therefore, I recommend this manuscript for publication after minor revision.

Detail comments:

  1. In Figure 4b, what is the (10) lattice plane? I suggest that the authors should confirm it.
  2. How many experiments did the error bars come from? I suggest that the authors should provide it in the revised manuscript.
  3. What about the structure of 4.3% polydopamine after cycles?
  4. Some important literatures involving 3D nanocarbon for energy storage need to cited in the revised manuscript (10.1016/j.nanoen.2018.08.075; 10.1021/acsami.9b02060; 10.3390/nano11051130).

Reviewer 2 Report

Zhu et al. studied the fabrication of three-dimensional nanocarbon with improved yield and capacitive performances. The work is interesting and can be considered for acceptance after minor revisions. 

1. Typical TEM results are suggested to provide to better describe the detailed characteristics of as-obtained porous nanocarbons.

2. SEM images of electrodes before and after the electrochemical cycle are suggested to provide in the revision.

3. Figure 3. The type of the hysteresis loop should be stated.

4. It is suggested that the authors compare the electrochemical properties of the materials with those of previously reported materials. An additional table is suggested to provide.

5.  Some references are too old, it is suggested to cite the latest achievements related to biomass-derived porous carbons in recent years.
[1] Journal of Colloid and Interface Science 2020, 569, 22–33. DOI: 10.1016/j.jcis.2020.02.062.
[2] Journal of Colloid and Interface Science 2022, 606, 817–825. DOI: 10.1016/j.jcis.2021.08.042.
[3] JOURNAL OF ALLOYS AND COMPOUNDS 2022, 892, 162129. DOI: 10.1016/j.jallcom.2021.162129.
[4] JOURNAL OF ALLOYS AND COMPOUNDS 2021, 885, 161014. DOI: 10.1016/j.jallcom.2021.161014.

Reviewer 3 Report

Recommendation: minor revision.

Comments: In this manuscript, the authors fabricated a series of polydopamine doped cellulose nanofiber-derived three-dimensional nanocarbon via a facile pyrolysis strategy as a free-standing electrode for supercapacitor. The morphology and electrochemical performance of these metal three-dimensional nanocarbon are good.  The experiment data relevant to these three-dimensional nanocarbon electrodes offered in this manuscript are sufficient to support the conclusion. Therefore, I recommend that this manuscript can be accepted with minor revision. Some issues that need to be further settled are as follow:

  1. In the introduction, the authors better describe the advantage of these binder-free electrodes and the roles polydopamine played in the energy storage field. Some related articles may be referred to in this section, such as Eng. J., 2020, 387,124061; Nanoscale, 2019, 11, 23110-23115; Chem. Eur. J., 2020, 26, 14708-14714. Nano Energy, 2015, 11, 366-376. J. Mater. Chem. A, 2016, 4, 13352-13362.
  2. The thickness of these three-dimensional nanocarbon electrodes is the better batter offered in the manuscript.
  3. To demonstrate the polydopamine is doped or only loaded on the surface of the cellulose-derived nano carbons, the SEM of these electrodes after the test is better to offer in the manuscript.
  4. These three-dimensional nanocarbon electrodes are flexible? Can it show in some flexible mode?
  5. The authors should compare their electrochemical performance with other recently reported carbon-based materials.
